# The COVID-19 pandemic eroded system support but not social solidarity

**Alexander Bor** [1,2]*, **Frederik Jørgensen**[2], **Michael Bang Petersen**[2]

**1** Democracy Institute, Central European University, Budapest, Hungary, **2** Department of Political Science, Aarhus University, Aarhus, Denmark

* bora@ceu.edu

**Data Availability Statement:** All relevant data are publicly available from the OSF repository (https://osf.io/fwy5j).

**Funding:** This research has been supported by the Carlsberg Foundation's grant CF20-0044 awarded

## Abstract

While the World was busy mitigating the disastrous health and economic effects of the novel coronavirus, a less direct, but not less concerning peril has largely remained unexplored: the COVID-19 crisis may have disrupted some of the most fundamental social and political relationships in democratic societies. We interviewed samples resembling the national population of Denmark, Hungary, Italy and the US three times: in April, June and December of 2020 (14K observations). We show that multiple (but not all) measures of support for the political system decreased between April and December. Exploiting the panel setup, we demonstrate that within-respondent increases in indicators of pandemic fatigue (specifically, the perceived subjective burden of the pandemic and feelings of anomie) correspond to decreases in system support and increases in extreme anti-systemic attitudes. At the same time, we find no systematic trends in feelings of social solidarity, which are largely unaffected by changes in pandemic burden.

## Introduction

The COVID-19 pandemic is the first once-in-100-year pandemic to hit modern, highly developed, individualized democracies. Before its outbreak, political and health experts have consistently warned that pandemic outbreaks "could cause (. . .) social disruption on a massive scale" [1], hold "the potential for serious social disruption" [2], and had "caused significant political and social disruption" in the past [3]. These warnings reflected in part that many Western democracies were already strained by political tensions prior to the onset of the COVID-19 pandemic [4, 5]. On the basis of lessons of the 1918 pandemic, Parmet and Rothstein [6, 1435] warned that "in our era of political polarization, 'fake news,' and tribal politics, [the fading] trust in the media, government officials, and even science . . . can be catastrophic if a [new] pandemic arises."

With the COVID-19 pandemic residing, it is now possible to evaluate the wider political disruption following in its wake. While COVID-19's disastrous effects on public health and economic activities are well-studied, surprisingly little attention has been paid to the pandemic's political impact on democratic societies. Specifically, we examine whether the pandemic disrupted citizens' views of (a) each other and (b) the political system. Thus, we focus directly

to MBP. The funders had no role in study design, data collection and analysis, decision to publish, or preparation of the manuscript. https://www.carlsbergfondet.dk/en

**Competing interests:** The authors have declared that no competing interests exist.

on the two major forms of relationships in any society: Horizontal relationships of solidarity between society's members and vertical relationships of power and legitimacy between people and the state.

Understanding this impact is of key importance. During the pandemic, citizens who trust each other and the political system comply more with advice regarding physical distancing, testing [7] and vaccination [8, 9]. Deteriorating social solidarity and political legitimacy may thus have made pandemic management more difficult as the crisis prolonged. Furthermore, the solution of a number of other monumental social challenges–such as the climate crisis–are highly dependent on the public's willingness to engage in and support collective action, which again requires social and political trust [10]. A negative impact of the pandemic can thus make the handling of these other crises more difficult for societies. Finally, many democracies have been experiencing increased instability over the last decades [5]. If the COVID-19 pandemic fueled this development, democratic societies may face significant turbulence in the coming years.

We ask whether the pandemic eroded social solidarity and system support across democracies. Answering this question is demanding in terms of causality and generalizability. Under the circumstances, we provide what is one of the strongest possible empirical designs: A multi-wave multi-country panel survey with pre-pandemic benchmarks, which allow us to study citizens' attitudes across democratic societies as the crisis unfolded. Specifically, we collected data in the first critical phase of the pandemic–between its onset and the roll out of vaccines–from four countries: the United States, Italy, Denmark, and Hungary. These are all democracies but differ both in how severely they have been affected by the pandemic and in their wider level of social conflict. If we find that the pandemic impacts these diverse countries in similar ways, we should expect this to generalize to the broader population of Western democratic countries.

## Vertical and horizontal relations under strain

Existing research has focused primarily on the initial impact of the pandemic in the Spring of 2020. Studies found that decisive government action against the coronavirus led to a rally around the flag, increasing support for national leaders, trust in the government and satisfaction with democracy [11, 12]. Evidence from Germany indicates that these initial boosts to trust have been larger in areas with a more severe health threat, but did not depend much on the specific policy response (possibly because of the relatively low exposure levels at the time). Individual level data from the Netherlands provides a plausible psychological mechanism for these effects: anxiety due to the local surges in COVID-10 infections lead to citizens to "rally around their political institutions as a lifebuoy" [13, 1007]. Kritzinger and colleagues [14] identified similar dynamics in Austria, but also documented two worrisome trends: first, trust in the Austrian government approached normal levels a few months following the onset of the pandemic. Second, the surge in trust was much more modest in France, where pandemic response was more polarized. Indeed, the rally-around-the-flag effect is not automatic. It critically depends on beliefs about the democratic institutions' capability to handle the crisis [15]. As the crisis prolonged, increasing doubts about this capability may have emerged.

The World Health Organization (WHO) has warned about "pandemic fatigue" [16] and a "global epidemic of misinformation" [17], both highlighting tensions between citizens and authorities. In a study of 144 countries during the COVID-19 pandemic, researchers documented violations of democratic standards in over 90% of the cases, including many democracies [18]. Consistent with this, 2020 saw a seven percent increase in public protests worldwide, many of them directly targeting pandemic restrictions [19]. Experimental evidence demonstrated that COVID-19 induced anger–but not fear or anxiety–depresses support for democracy and increases support for authoritarian alternative [20, 21]. At the same time, democratic

standards themselves may be waning: both American and British respondents supported emergency measures encroaching civil liberties proposed by their party or by trusted experts [22].

While these accounts highlight how the pandemic has tested the vertical relationships between citizens and the political system, there is also evidence that the pandemic has strained horizontal relationships, or in other words, social solidarity. Survey research, for example, documents widespread shaming and blaming of fellow citizens who do not comply with health advice [23]. Conflicts arising from such moralistic condemnation may undermine interpersonal trust between citizens. The global spread of the coronavirus may have also made citizens more suspicious of foreigners. Tightening border security has proved a popular policy even in EU countries that otherwise cherish free movement of citizens [24]. Meanwhile, hate crimes against Chinese minorities have increased in response to the pandemic [e.g. 25].

Finally, both the disease of COVID-19 itself and the restrictions implemented to contain the virus affect marginalized groups disproportionally [26]. Such inequalities raise the importance of helping the disadvantaged through redistribution. Moreover, as inequalities grow, they can become a hotbed of grievances, not only among those suffering, but–perversely–also among privileged groups, who may think that the pandemic is none of their concern, or worse, that their own suffering is caused by the non-compliance of the marginalized. Feelings of social injustice and group-based grievances, in turn, are important antecedents of social conflict [27].

While these observations may suggest that the pandemic has strained both the horizontal and vertical relationships in democratic societies, we lack a systematic assessment that integrates generalizable, cross-country comparisons with temporal comparisons about the state of these relationships prior to the pandemic and as the pandemic unfolded. We also lack an understanding of the psychological sources of the potential strain.

From a psychological perspective, the pandemic's peril lies in its potential to disrupt people's sense of normality, which may result in feelings of low power and a lack of control. Research on social change and populism has documented how these feelings lead to both "vertical and horizontal opposition" [28, 125], fueling support for right-wing populism, elite discontent and prejudice [29]. Part of the psychological mechanism may include that other groups in society as well as the political system constitute tangible enemies compared to diffuse feelings of distress. Opposition towards these targets may serve as a way to restore a sense of control. The slogans of successful populist movements such as Trump's "Make American Great Again" and the Brexiter's "Take Back Control" speak directly to this need [28].

It is likely that the COVID-19 pandemic elicited both distress and feelings of low control. For many, the pandemic created the most severe social upheaval of their lifetime. The disease and the restrictions used to contain it (e.g., lockdowns) disrupted people's regular social habits and prompted physical distancing, which over extended periods of time may harm mental and physical health [30], and generate a sense of "fatigue" [4, 16]. In addition to the distress from social isolation, people over the pandemic have been worried about their health and economic prospects and expressed concerns about violations of fundamental rights and freedoms [31]. This compound of stressors, and the associated pandemic-related distress, "may lead to anger at those perceived as causing it" [32], whether this is the authorities imposing restrictions [19] or groups spreading the virus [33].

On this basis, we investigate whether the COVID-19 pandemic eroded people's *social solidarity* and people's *system support*. We define social solidarity as citizens' beliefs that other citizens deserve help when in need, and can be generally trusted. We define system support as citizens' approval or disapproval of the political system they live in–in our case, democracy. As potential psychological mechanisms of fatigue, we assess 1) a broad, compound measure of various burdens inflicted by the COVID-19 crisis, and 2) a narrow measure of the perceived

loss of control over one's life. Given the severity of the crisis, we go beyond traditional measures of system support and also assess whether these psychological mechanisms drive *extreme discontent* in the form of support for destruction, populist sentiments, the sharing of misinformation and anti-social mindsets.

## Methods

### Data

We believe, our study provides one of the strongest possible empirical designs to study the attitudinal effects of the COVID-19 crisis: our data are comparative, timely, reflective of national populations, measures people at multiple times and can be benchmarked to pre-pandemic values. We discuss each of these merits in turn below.

We collected survey data from four countries, which constitute a diverse selection of Western democracies: Denmark, Hungary, Italy and USA. To the extent that we can identify trends that are similar across these four countries despite their differences, we may assume that similar processes characterize other democratic societies. Our countries vary on the level of democracy (highest in Denmark, lower in Italy and the USA, and lowest in Hungary), political polarization (low in Denmark, high in Italy, very high in USA and Hungary), violations of democratic standards in handling the pandemic (no violations in Denmark, minor violations in Italy, moderate violations in the US and Hungary), GDP (lowest in Hungary, medium in Italy, highest in Denmark and USA), government effectiveness (very high in Denmark, lower in USA, lowest in Hungary and Italy), and health equality (very high in Denmark and Italy, lower in Hungary, lowest in USA). Section B.1 in the S1 File reports detailed statistics underpinning these differences.

We started data collection in April 2020 within weeks of the WHO's announcement of COVID-19 as a pandemic on March 11. At the time, Italy was suffering from a massive epidemic toll and severe lockdowns, Hungary and Denmark were locked down but the epidemic was kept under control; meanwhile in the US most places were relatively open, but the epidemic was gaining speed. Our study design reflects the massive uncertainty surrounding the pandemic at the time. During these early weeks, it was unclear how much disruption the pandemic would cause, for how long, and on which areas of life. Accordingly, our ambition was to tap into a wide variety of social and political attitudes, which are explained in more detail below.

Data collection was performed by YouGov survey agency. Participants provided informed consent, had the opportunity to express concerns, and were re-numerated for their time via the survey agency. YouGov recruited participants from their online panels by quota sampling on age, gender, geography, education, and in the US also race. Accordingly, our initial samples resembled the marginal distribution of these variables in the population. Given that traditional face-to-face interviews which could yield more representative samples were not feasible (or ethical) due to the lockdowns, we believe these quota-sampled online surveys offer the highest quality and safest data under the circumstances. Our initial sample sizes were set to 1500 respondents per country with the goal of retaining samples of 1000 respondents by wave 3 assuming a 33% attrition (which has proved overly optimistic in the USA and especially Hungary). No a-priori power analyses were conducted, sample size was based on monetary constraints. Our total sample size in Wave 1 was 6,131.

Given our ambition to study the social and political damage of the unfolding COVID-19 crisis, we employ a panel setup re-interviewing the same participants in two additional waves. We invited participants to take the same survey a second time in June 2020. This decision has been motivated by two considerations: first, by that time the first wave of the pandemic has receded in all of our countries (although this meant different infection rates in different

places); and second, the Black Lives Matter protests and counter-protests in the United States indicated that considerable social tension had accumulated. The total sample size in Wave 2 was 4,568 (A small subset of data from the first two survey waves is also reported by Bartuse-vičius and colleagues [19]).

We fielded a third and final wave in early December of 2020 re-inviting all participants from wave 1, even if they had missed wave 2. 4,018 respondents participated in wave 3. By this time, the second pandemic wave was peaking in most countries, matching or–in the case of Hungary–superseding the first wave's levels of infection. Our final wave also coincides with the roll out of the first vaccines against the novel coronavirus. Thus, our study offers important insights for the first phase of the pandemic, hallmarked by reliance on physical distancing and other government mandated public and social measures to contain the virus.

All in all, we collected 14,717 observations from 6,131 individuals. 3,620 respondents participated in all 3 waves (1,080 from Denmark, 639 from Hungary, 1,002 from Italy and 899 from the USA) yielding a balanced panel sample of 10,860 observations. Section B.2 in S1 File demonstrates that our balanced panel sample shows only minor deviations from population margins used as demographic benchmarks. We adjust these minor deviations using entropy balancing.

## Outcome measures

Our survey is unusual in that it includes fourteen outcome variables. Given the high uncertainty surrounding the potential socio-political effects at the onset of the pandemic, our strategy was to paint a broad overview of respondents' perceptions of society and politics. We bundle outcomes into three broad categories, consistent with our theory: social solidarity, system support and extreme discontent. We do not claim that the measures bundled together form latent traits, just that they show sufficient theoretical and empirical consistency that it is helpful to consider them together. For transparency, we report all results both for individual measures and pooling by category. All measures within social solidarity and system support are based on validated items from the European / World Values Survey. In Section A in S1 File we report full question wordings, while in Section B in S1 File we report descriptive statistics, test-retest correlations, and reliability estimates for our outcomes.

**Social solidarity.** The four variables under social solidarity concern horizontal relationships between citizens. Did the pandemic change how much people trust and support other people in society? First, we measure *support for redistribution* with three items, which prompt respondents to position themselves on a 10-point scale whose endpoints are marked by two statements, e.g. "Individuals should take more responsibility for providing for themselves (1)" or "The state should take more responsibility to ensure that everyone is provided for (10) (Due to a coding error, answers to one of the three solidarity items had to be discarded from the first wave of the Danish survey)" Second, we measure *attitudes towards immigrants* with similar 10-point scales, e.g. "Immigrants are a strain on a country's welfare system (1)" versus "Immigrants are not a strain on a country's welfare system (10)" We scale items such that higher values indicate more solidarity with immigrants. Third, we measure *social trust* with a binary item recording response to the question "generally speaking, would you say that most people can be trusted or that you can't be too careful in dealing with people?". Finally, we measure *attitudes towards surveillance* as an indirect measure of fellow citizens' trustworthiness. Here, we use three items asking whether the government should have the right to e.g. "to collect information about anyone . . . without their knowledge". Answers to a four-point scale are reversed to indicate a rejection of state surveillance and thus indicating trust in fellow citizens.

**System support.** The four variables under system support concern vertical relationships between citizens and the state. Did the pandemic change how much people approve the political system they live in? Importantly, this concerns (diffuse) support for the broader institutional order than (specific) views about the current government [CF 34]. First, we measure *satisfaction with the political system* with a question directly prompting how satisfied respondents are with "how the political system is functioning these days." Second, we rely on a similar question asking "how democratically is this country being governed today" to tap into perceptions about the *level of democracy*. Third, to measure *support for democracy*, we combine three items about respondents' impressions on how good or bad various types of political systems would be for their country: a democratic political system, a strong leader who does not have to bother with parliament and elections, and an army rule. We flip the latter two items, thus higher values on the resulting index indicate higher support for democracy (This battery includes a fourth item about "having experts, not government, make decisions according to what they think is best for the country". Given most politicians' limited knowledge of virology or epidemiology, it is unclear to what extent a technocratic approach to decision making during the pandemic threatens democracy. Therefore, we exclude this item from the index). Finally, to assess more diffuse feelings towards the nation state, we asked respondents on how *proud citizens* they are on a 4-point scale.

**Extreme discontent.** The scales tapping into extreme discontent go a step further in measuring respondents' potential frustrations. Beyond a mere lack of social solidarity or system support, they ask if people also entertain more radical attitudes. Freed from the burden of relying on measures which could be benchmarked against pre-pandemic surveys, these variables also rely on more items, reducing measurement error. To measure radical anti-systemic attitudes, we use the 8-item *Need for Chaos* scale (e.g. "I think society should be burned to the ground") [35]. To measure *populism*, we use a six-item scale, e.g. "The government is pretty much run by a few big interests looking out for themselves" [36]. Finally, we also asked respondents to indicate to what extent they would *believe or share a false news* headline stating that "The coronavirus has been developed intentionally in a lab to be used as a bioweapon". Respondents used standard seven-point Likert scales to indicate their agreement or disagreement with each item. All indices are recoded such that higher values indicate more extreme discontent.

Finally, three unbenchmarked additional measures, for affective polarization, perceived financial prospects and support for the government's handling of the coronavirus crisis are reported in Section C.7 of the S1 File.

## Predictors

Our ambition is to estimate the individual level effects of the COVID-19 crisis. But how to quantify a pandemic? One approach could have been to rely on objective indicators, like the incidence of COVID-19 to measure the severity of the pandemic, or stringency indices to measure the severity of the government's response to the pandemic etc. Yet, these measures would be both too broad–in the sense of lumping too many people together despite markedly different personal experiences–and too narrow–in the sense of zooming in on a specific effect of a global crisis. Therefore, instead we rely on two subjective measures of the COVID-19 crisis' immediate impact on respondents. First, we designed an original scale that taps into four (The original battery included a fifth dimension, government evaluations, which could be seen as endogenous to some of our outcomes. Therefore, our main analyses do not include this factor, but we replicate our findings with the full battery in the Section D.1 of the S1 File) dimensions of the subjective burden of COVID-19 crisis: health, finances, social connection and anomie.

Our aim with this novel measure, therefore, is to create a battery reflecting that the COVID-19 crisis is a compound treatment, affecting several domains of citizens' lives. Each of the four dimensions is measured with two items, one positively and one negatively worded. For example, the two items about finances read "The coronavirus crisis has affected negatively my financial situation" and "My finances are in good order despite the coronavirus crisis."

Second, we rely on a validated 17-item scale of *anomie*–or meaninglessness–developed by Yang [37]. Zooming in on feelings of anomie provides important additional insight beyond a broad measure of covid burden for two reasons: (1) previous research found that anomie underlies a wide range of anti-social attitudes and behaviors from radical anti-systemic attitudes [35]; (2) it has been argued from the first weeks of the government mandated lockdowns that the severe social isolation imposed on societies may easily contribute to feelings of anomie. Indeed, multiple items of our psychometrically validated anomie-scale–e.g. "I live a trapped life" and "I have no control over my destiny"–offer an uncannily insightful description of life under a quarantine. We form simple additive indices from both of these scales indicating higher burden and anomie, respectively.

In Section C.3 in S1 File, we offer both within-respondent and cross-sectional analyses on who feels more burden and anomie. These show that anomie and burden are influenced slightly, but not substantively by changes in personal health, getting infected with COVID-19, or changes in employment status. We also find that young respondents, and those with lower education are more vulnerable to both covid burden and anomie. Covid burden and anomie are related to each other both conceptually (anomie is one of the four facets of the burden index), and empirically, with a Pearson's correlation of $r = 0.56$.

## Modeling

Our paper has two ambitions. The first is descriptive: how much solidarity did people feel towards their fellow citizens? How much did they support the political system they lived in? How much extreme discontent did they express? How much burden and anomie did people feel at various stages of the pandemic? Monitoring large samples of people living in four different countries, our data is uniquely well-suited to answer these questions. Whenever possible (Benchmarks for 5 items in the US are not available: one of three items in the support for redistribution scale, and four of six items in the tolerance for immigrants scale. Thus, these two scales are formed using the two benchmarked items in the US data. We use the full batteries in the within-respondent analyses), we benchmark our measures against the last pre-pandemic data collection of the European Values Survey (Denmark–late 2017; Hungary–first half of 2018; Italy–late 2018) and the World Values Survey (USA–Spring 2017). Specifically, we transform our individual level responses to z-scores using the weighted mean and standard deviation from the benchmark survey. To minimize the potential impact of difference in data collection and sampling, we used entropy balancing [38] to re-weight both benchmark and original surveys to the same population margins applied to our own data (age, gender, education and region); see Section B.2 in the S1 File for distributions in demographic variables for both weighted and unweighted, full and balanced panel samples. In our descriptive analyses of the outcome variables, we rely on all responses collected, irrespective of whether they come from respondents who later dropped out or not (total N = 14,717). This decision increases the precision of our estimates in earlier waves without affecting our substantive conclusions.

Our second ambition is to estimate the effects of COVID-19 burden and anomie on our outcome variables. As we are unable to identify or to experimentally induce exogenous shocks in burden and anomie, our causal ambitions must be tempered. Yet, our panel data has a large advantage over standard cross-sectional surveys, by allowing us to zoom in on within-

individual changes. In other words, instead of comparing people with high burden to people with low burden (while making futile attempts to control away the several other differences between these two groups), we are comparing people to themselves contrasting times when their burden was higher to times when it was lower. More technically, we rely on linear regressions with respondent and country-wave fixed effects, also known as two-way fixed effects (2FE) models. These fixed effects demean all our measures and thus respondent $i$'s response at time $t$ changes its meaning to a deviation from what we would have expected given the respondent's average answers (across all waves) and the tenor of the time (across all respondents). These models purge all selection bias from confounders that are either respondent-specific but time invariant–*i.e.* affect a respondent in the same way across the three waves–or time-specific but respondent-invariant–*i.e.* affect all respondents in a country in the same way in a given survey wave. At the same time, these models remain vulnerable to bias from time-varying confounders and reverse causality. Yet, we believe these within-respondent analyses constitute a good (perhaps the best) compromise balancing internal validity and generalizability, especially because interventions experimentally manipulating the impact of the pandemic are difficult if not impossible to conduct.

We report two types of linear two-way fixed effects models: 1) standard 2FE models that regress each outcome variable (e.g. support for democracy) on a predictor (e.g. the perceived burden of COVID-19). At the same time, to reduce the complexity of our findings, we also average across the outcomes within each of the three categories by building pooled models. These models stack our data such that each respondent appears three or four times in our sample, once for each of the constituting outcomes within a given category. Here, we add a third fixed effect, namely outcome, thereby removing variation caused by the fact that people on average agree more with some scales than with others within a category. In practice, this pooling method yields identical results to simply forming an additive index across the three sets of outcomes for each individual, but its uncertainty estimates are a bit more conservative. All fixed effects models report standard errors clustered on respondents.

Because 2FE models rely exclusively on within-unit variance, it is important to scale effect sizes to realistic within-unit changes. After all, covid burden or anomie hardly ever change for a respondent from the minimum to the maximum value during our study period. Accordingly, we follow best practices and calculate the distribution of the (largest) within-unit changes in our predictors (see Figure OA7 in the S1 File) and identify a large but realistic within-unit change at the 95th percentile of this distribution [39]. Coincidentally, this value is close to a standard deviation of the overall (between respondents) variance in our sample for both COVID-19 burden (1.00) and anomie (1.18) reflecting the turbulence of the study period. This value corresponds to roughly twice the average largest within unit-change between the three waves (burden = 0.48; anomie = 0.51), but it is still half, or a third of the maximum observed within-unit changes in our samples (burden = 2.05; anomie = 3.36). Finally, we note that in these analyses we rescale all outcomes to 0–1 range. To sum up, the coefficient estimates we report denote the expected *percentage point change* in the outcomes corresponding to a large but realistic within-unit change in the predictor.

## Open practices statement

The data, command scripts, and questionnaires necessary to reproduce and replicate our analyses are available at https://osf.io/fwy5j/.

## Results

Fig 1 displays descriptive trends in horizontal relationships of solidarity and vertical relationships of system support. As noted, we can benchmark these measures against pre-pandemic levels (trends in unbenchmarked extreme discontent measures are reported in Section C.5 of the S1 File). Colors denote individual outcome measures with black reserved for the pooled estimate. Recall that negative values indicate an average decrease compared to the benchmark. The y-axis is scaled to the standard deviations observed in the benchmark survey.

*How did social solidarity change throughout 2020 and compared to the pre-pandemic benchmark*? Pooling across variables and countries (black line leftmost facet), social solidarity remained remarkably similar in our sample compared to the benchmark surveys. We observe notable decreases in tolerance towards immigrants among Danes and Americans, as well as support for redistribution among Americans (in the range of -25% to -44% of a standard deviation). These negative changes are, however, offset by smaller opposite trends in several other variables. Indeed, when we look at the country averages (black lines within each facet), we see that by April social solidarity slightly increased in Hungary and Italy (by 10–14%), while decreased in Denmark and the US (by 5–14%). Furthermore, we find no evidence that social solidarity has been eroding throughout 2020.

*How did system support change throughout 2020 and compared to the pre-pandemic benchmark*? System support paints a markedly different picture than social solidarity. To begin with, already by April, system support decreased by 26% of a standard deviation averaging across countries and outcomes (black line, leftmost facet). Moreover, this trend continued throughout the corona-crisis. While the summer relieved a bit the burden of the pandemic, system support continued to decrease to about -32% by July and to -37% by December. The severity of these trends is well summed up by the fact that of the 16 estimates in wave 1 (four outcomes in four countries) only three increased compared to the benchmark, and both of these dropped below the benchmark levels as the pandemic dragged on. Indeed, Danes' perceptions on the level of democracy decreased by an astonishing 92% of a standard deviation by December. A detailed analysis of all time trends (Section C.1 in S1 File) demonstrates that the changes are statistically significant at conventional levels in all three system support variables showing a negative trend (level of democracy, pride in citizenship, satisfaction with the political system), and that they are present in all four countries using the pooled estimates.

As argued, it is likely that these aggregated changes are driven by individual-level changes in the psychological burden or anomie triggered by the pandemic. We first assess the overall changes of these psychological factors as the pandemic unfolds.

*How did the subjective burden of COVID-19 and anomie change throughout 2020*? Fig 2 displays the descriptive trends in our two independent variables, covid burden (yellow) and anomie (green), and an additional single-item measure of anomie that we could benchmark to pre-pandemic levels (purple). The plot also displays the running average of COVID-19 related deaths standardized to population size (Data downloaded from the European Centre for Disease Prevention and Control website on 2021-06-23. Link: https://www.ecdc.europa.eu/en/publications-data/data-national-14-day-notification-rate-covid-19). The plot highlights that despite notable differences in pandemic trends across the four countries, the trends in people's subjective perceptions were remarkably similar. In each country, people experienced a high burden at the onset of the pandemic in April. Our benchmarked anomie measure indicates that in all four countries, during the pandemic people felt that "what they do has no real effect on what happens to them" more than before the pandemic (average increases between 30 and 40% of a standard deviation. See details in Section C.10 of the S1 File). Compared to this initial change, fluctuations during the pandemic are much smaller. Burden and anomie decreased a

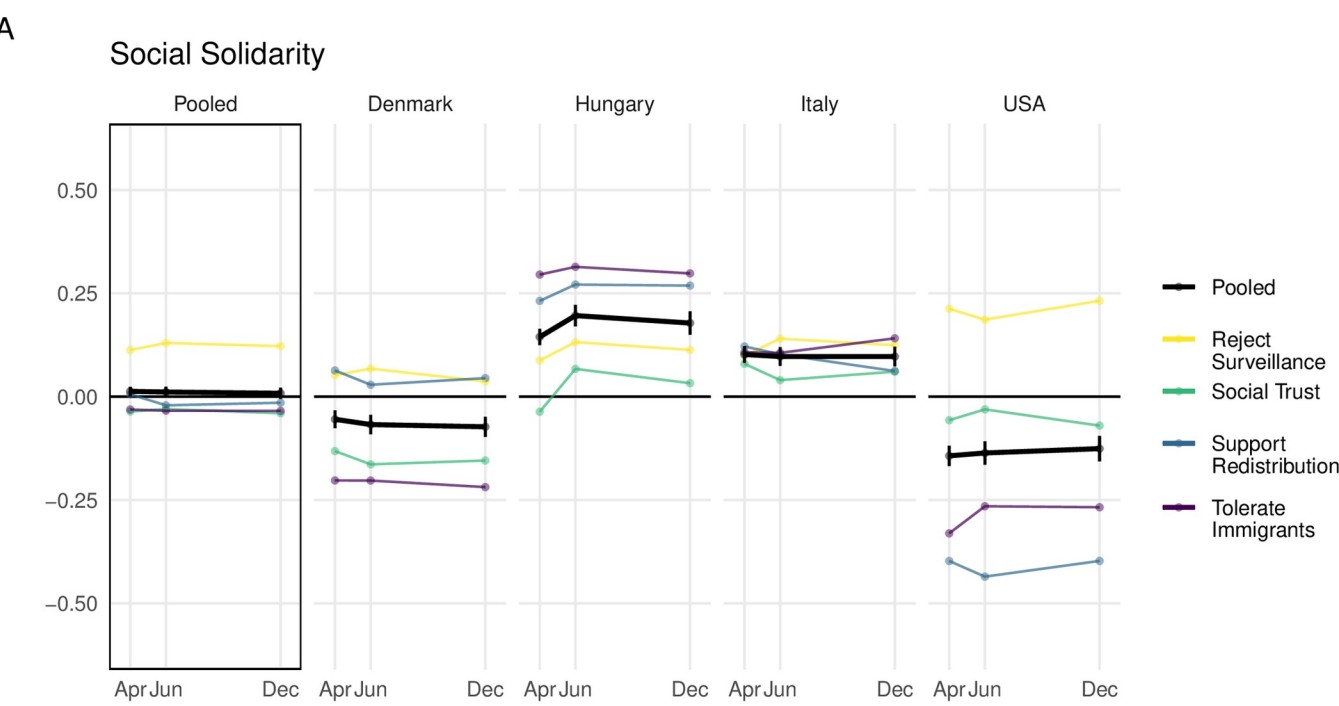

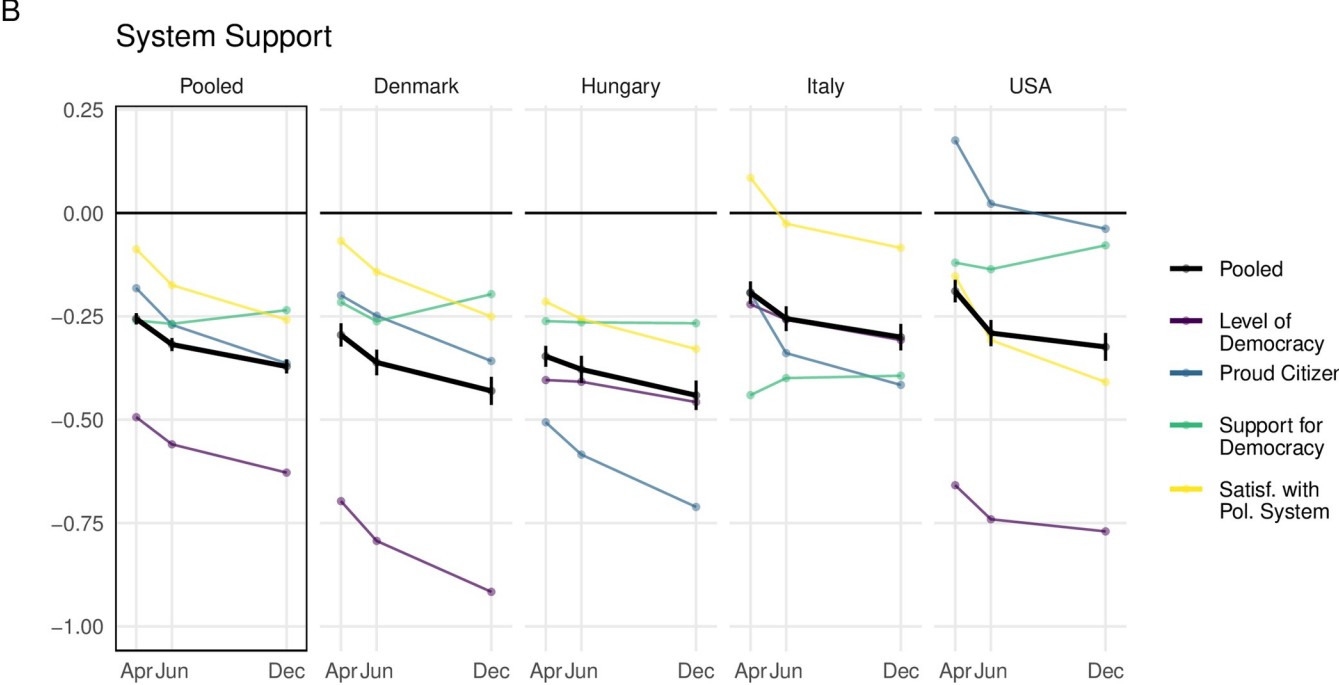

**Fig 1. Descriptive trends in social solidarity and system support benchmarked against pre-pandemic baselines.** Values are z-scored to nationally representative surveys conducted pre-pandemic by World and European Values Surveys. Negative values indicate a decrease compared to the benchmark. The scale refers to standard deviation changes. The black lines pool across the outcome variables. The leftmost facets pool across the four countries. The plot shows a sizeable and growing decrease in most measures of systemic support, whereas the changes in social solidarity are smaller and do not appear to change throughout the study period. See Section C.1 in S1 File for more details and statistical tests.

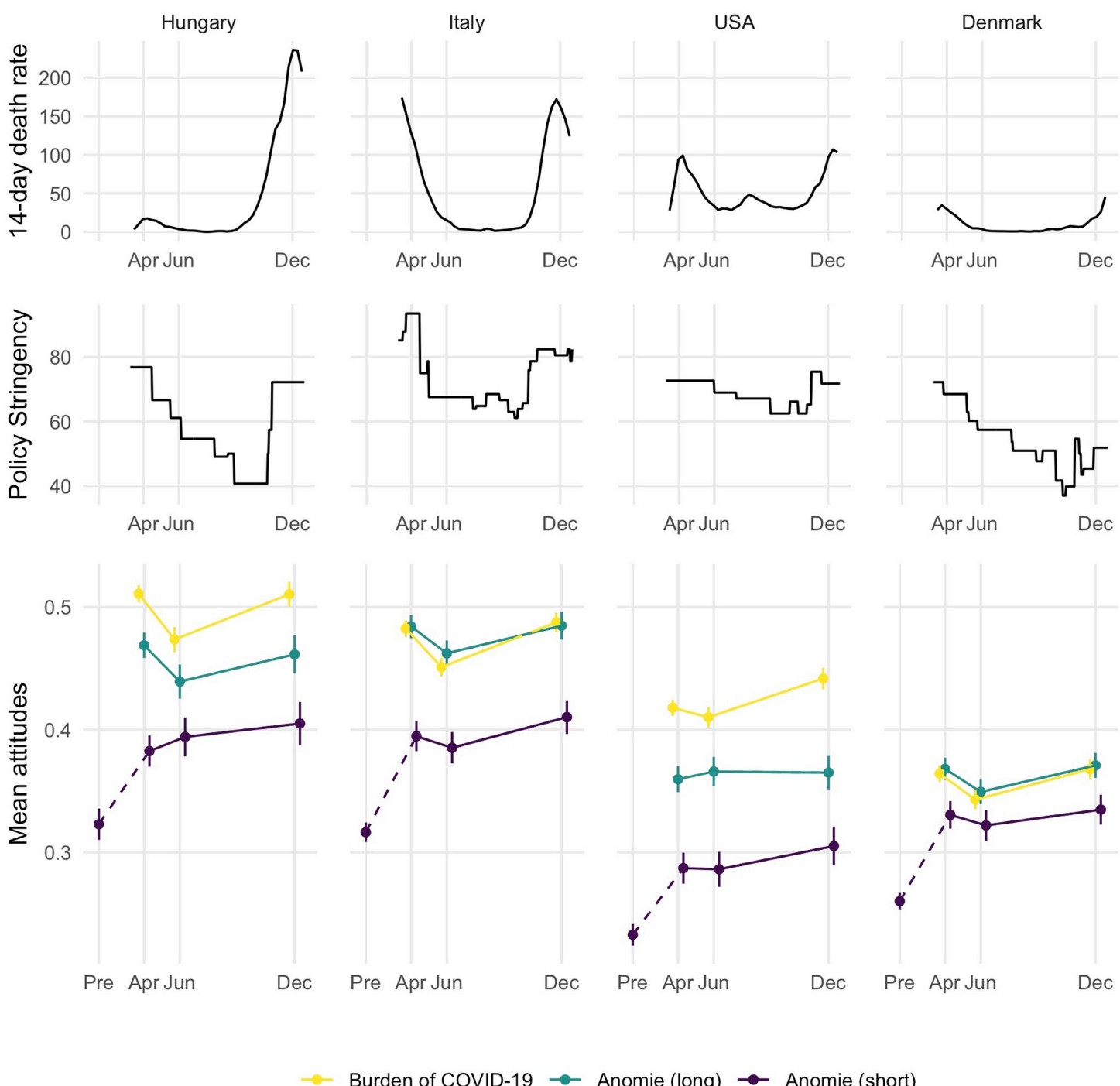

**Fig 2. Country-level descriptive trends during the study period.** COVID-19 deaths per million people from the European CDC (top panel), stringency of government response from the Oxford Covid-19 Government Response Tracker (middle panel), and average perceptions of the burden of COVID-19 (bottom panel) based on our data. "Pre" denotes values from the pre-pandemic benchmark surveys for the short anomie measure (2017–8).

bit in July, but bounced back by the end of the year. Cross-national differences in the levels of perceived burden and anomie roughly track the differences in pandemic severity with Hungary and Italy having both higher death rates and burden; Denmark having both low death rates and burden; and with the US in the middle on both.

At the same time, Fig 2 also offers a reminder that a subjective burden is not perfectly in tune with objective circumstances. Hungarians express as high a burden in April, when the pandemic was kept well under control, than in December, when it certainly was not. No doubt, the severe uncertainty and the harsh restrictions made many respondents miserable early on, which even the high death toll of the second wave could not surpass. Similarly, despite deaths falling to low levels and most governments lifting the harshest of restrictions by June, burden and anomie fell only by a few points on average. It is also noteworthy that while system support exhibits a trend of slow decay over time, perceptions of burden and anomie fluctuate more with the objective circumstances, suggesting that the potential effect of these subjective assessments on system support involves some complexity. We return to this key observation in the discussion.

Next, we ask whether these changes in subjective perceptions about the pandemic are associated with changes in perceptions of horizontal and vertical societal relationships. Therefore, now we turn to our linear two-way fixed effects models exploring whether changes in burden or anomie relate to changes in our outcomes. Fig 3 displays the regression coefficients for each group of outcomes and independent variables pooling across the four countries.

*Is an increase in the subjective burden of COVID-19 or anomie associated with a decrease in social solidarity*? The top left panel in Fig 3 indicates that covid burden is largely independent of social solidarity. Judging from the pooled DV, the average association between changes in burden and changes in social solidarity are very close to 0 and estimated very precisely (95% CIs: burden = [-0.01, 0.01]; anomie = [-0.02, 0]). This average within-individual association appears to be the result of a small positive relationship with support for redistribution, and small negative relationship with tolerance for immigrants and social trust.

The top right panel in Fig 3 shows that the associations of anomie are parallel to those of covid burden: changes in anomie were not associated with systematic changes of attitudes related to social solidarity on average. The only divergence between the two independent variables is the opposite trends in attitudes towards surveillance. An increase in anomie is associated with a slight decrease in the rejection of surveillance, although we do not find the same pattern for changes in burden.

*Is an increase in the subjective burden of COVID-19 and anomie associated with a decrease in system support*? Yes. The middle row of panels in Fig 3 demonstrates that when respondents' covid burden or anomie increase they also tend to feel lower support for the political system. Pooling across the four outcome variables, a one unit increase in our independent variables is associated with a 3–4 percentage points decrease in system support (95%CIs: burden = [-0.04, -0.03]; anomie = [-0.04, -0.02]). This trend describes all four outcomes well: all our estimates fall between 2.5 and 5 percentage points.

*Is an increase in the subjective burden of COVID-19 and anomie associated with an increase in extreme discontent*? Yes. Judging from the bottom row of panels in Fig 3 when respondents' covid burden or anomie increases, they also tend to feel 2–5 percentage points more extreme political discontent (95%CIs: burden = [0.01, 0.03]; anomie = [0.04, 0.06]). Again, this trend can be observed to some extent for each of our three outcome variables, yet there are more interesting heterogeneities. First, changes in anomie result in more than twice as large changes in extreme discontent compared to changes in covid burden. Second, we observe the largest shifts in Need for Chaos with an 8 percentage points change associated with a one unit change in anomie (95%CI: [0.07, 0.10]). Of these three outcomes, we observe smaller yet still

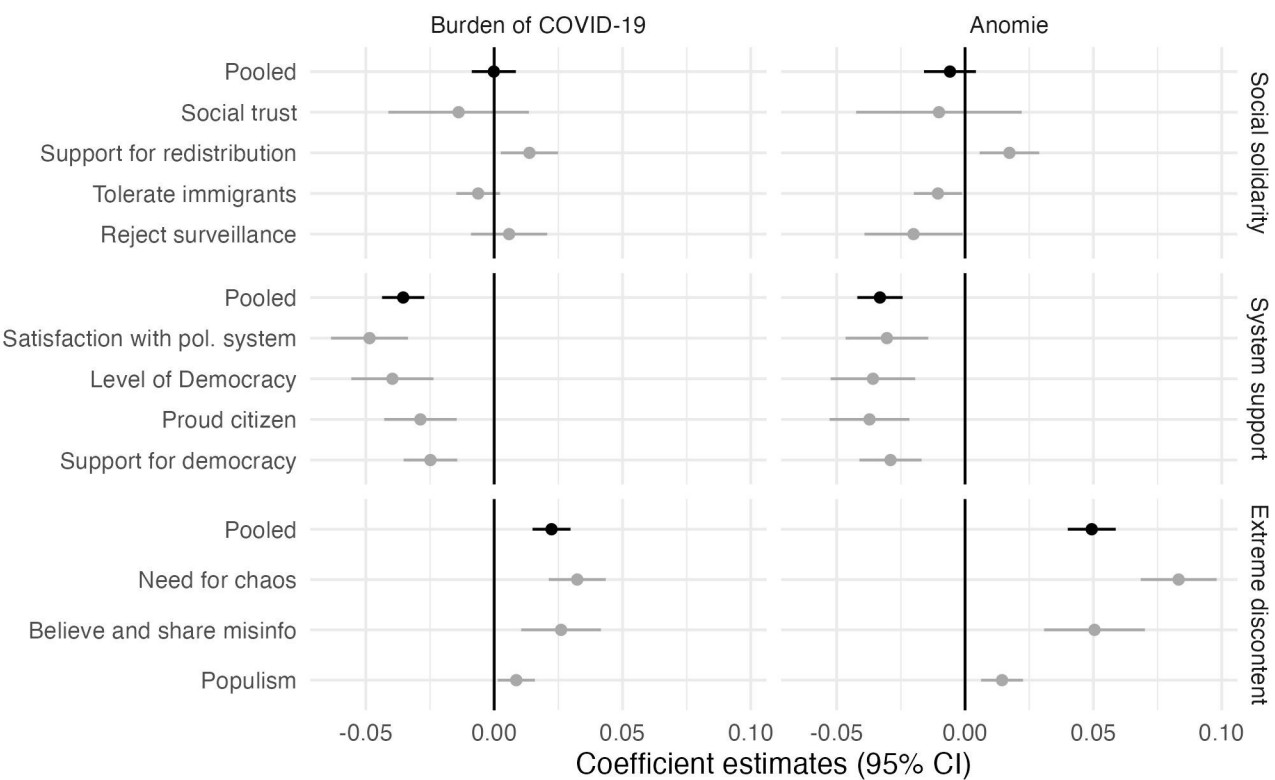

**Fig 3. The burden of COVID-19 and anomie are associated with a decreased system support and increased extreme discontent but not with changes in social solidarity.** The figure reports two-way fixed effect regression coefficients with 95% confidence intervals from models pooling across the four countries. The two columns denote our two *independent variables*. Each row is a separate dependent variable. Model details are reported in Tables OA8-OA13 in S1 File.

concerning trends in believing and sharing misinformation, whereas populist attitudes remain largely unaffected by changes in burden and anomie.

## Additional analyses and robustness tests

*What other attitudes do changes in covid burden and anomie predict*? Our surveys included three additional outcome variables, which reinforce some of the results presented so far (details in Section C.7 of S1 File). In line with the small and inconsistent associations with Social Solidarity, we find no evidence that changes in either covid burden or anomie are associated to changes in affective polarization. We do find however, that in line with the consistent negative associations with System Support, an increase in burden and anomie is also associated with a large decrease in support for the government. Finally, and unsurprisingly, we find that when respondents' burden or anomie increase, they also become more pessimistic about financial prospects.

*How much country-level variation do we observe in within-individual associations*? We replicate our within-respondent regression models independently for each of the four countries. These analyses demonstrate that for the most part the associations are remarkably consistent across countries, especially when looking at the pooled models (see details in Section C.6 of S1 File).

*Which factors of the subjective burden of COVID-19 drive the observed within-individual associations*? Curious about the relative role of the four facets of our measure of covid burden, we rerun our analyses separately using finance, health, anomie and social life. Figure OA11 in S1 File shows that when it comes to system support, feelings of anomie play by far the largest

role, although each of the other three factors show negative correlations as well. Surprisingly, the main correlate of extreme discontent appears to be burden on health, although–again– each of the other three factors show positive correlations too.

*How much confidence should we put in the internal validity of our estimates*? The linear two-way fixed effects models constituting the backbone of our analyses guard us against all (observed or unobserved) respondent- or time-specific confounders. At the same time, our analyses are vulnerable to bias from time-variant confounders or reverse causality (i.e., changes in the outcome feed back to future changes in the predictor). We perform two sets of robustness tests to assess the severity of these vulnerabilities to the internal validity of our estimates.

First, we rerun our models including leaded predictors. This means that we regress e.g. perceptions of Social Solidarity not only on perceived burden in the same wave, but also burden in the following wave. If the outcomes feed back on the predictors–i.e. change in the outcome precedes change in the predictor–we would expect to find substantively significant coefficients for the leads. We would also expect the inclusion of these leads to diminish the original estimates for the within-unit associations. Figure OA15 in S1 File reports results from these leaded models along the original estimates. Given the consistent null results with the variables under Social Solidarity, we focus on System Support and Extreme Discontent here. We find no evidence that associations with Extreme Discontent are affected by the inclusion of leads, all of which are close to 0. At the same time, we do find that multiple leads under the System Support category–including the pooled estimate–turn out to be statistically significant and in the same direction as the original estimates. In other words, when respondents' System Support decreases, they are more likely to perceive an increased covid burden–and to a lesser extent an increased anomie–in the following wave. This overall trend appears to be driven by satisfaction with the political system and level of democracy in the case of covid burden, and satisfaction with the political system and pride in citizenship in the case of anomie.

Second, we rerun our models including respondent-specific linear time trends. These models relax the parallel trends assumption by letting each respondent to follow their own linear trend, and seeks to find associations in the deviations over and above this trend. We note that with three panel waves, our estimates of these individual-specific trends are bound to be noisy, and thus results should be taken with a grain of salt. Yet, it is "heartening" [40, 178] that according to Figure OA16 in S1 File, our estimates remain substantively similar–if slightly diminished–with the inclusion of respondent-specific linear trends.

*Does system support measure anything beyond support for the government*? Yes. Support for the incumbent government is likely to influence and be influenced by system support. Nonetheless, our data underpins decades of political science research highlighting the distinct nature of system support and government evaluations [34]. The correlation between our pooled system support variable and government evaluations is medium ($r = .39$), ranging from $r = 0.01$ in the case of support for democracy to $r = 0.68$ in the case of system support. Moreover, when we rerun our individual-level models regressing system support variables on covid burden and anomie while adjusting for changes in support for the government, we find that the associations retain most of their size, 62% for burden, 78% for anomie (see Figure OA17 in S1 File).

As yet another empirical evidence underpinning that system support matters, in Section C.9 of the S1 File, we replicate and extend recent findings showing that those who perceive a heavy covid burden and high anomie have on average higher intentions to participate in political violence [19]. Here, we use data from Waves 2 and 3 to demonstrate that this relationship stands even in within-respondent analyses, which guard against all observed and unobserved time invariant confounders. In other words, when respondents' covid burden or anomie increases, they are also 3 percentage points more likely to have radicalism intentions (95% CIs

= [0.01,0.05]). Meanwhile, the association to changes in (non-violent) political activism intentions are smaller and do not reach statistical significance at conventional levels.

*Do we find similar substantive results in one-way respondent fixed effects models*? Yes. Recent methodological advances remind us that our interpretation of two-way fixed effects coefficient estimates–as associations between within-respondent changes beyond the common time shocks–hinges on the common assumption that the causal effects of burden and anomie are constant across units [41]. The critiques of two-way fixed effects models recommend verifying that more easily interpretable respondent-fixed effects models yield similar conclusions than two-way fixed effects models. Accordingly, we rerun our models with only respondent fixed effects to assess the relationships if we are unwilling to make these assumptions. The one-way respondent fixed effects models estimate the average associations between changes in predictors and outcomes within respondents across the sample. Figure OA18 in S1 File demonstrates that these estimates are in essence identical to our two-way fixed effects estimates reported above.

*Does zooming in on within-respondent changes (and omitting time invariant confounders) make a difference*? Yes. We also run one-way country-wave (i.e. time) fixed effects, which estimate the average associations between the predictors and outcomes *across* respondents in our sample. As these are essentially cross-sectional models, we adjust for standard demographic covariates: age, gender, and education. These estimates highlight the large benefits of our panel design: had we missed the opportunity to control for time invariant unobserved confounders, we would have grossly overestimated the associations between out variables. This pattern is especially striking in the case of social solidarity, where our within-respondent analyses consistently find negligible relationships, while the cross-sectional analyses (even with the covariates) find strong associations. For example, respondents with higher COVID-19 burden trust fellow citizens 14% points less than respondents with lower burden 95% CI = [-0.12, -0.15]. Meanwhile, those with higher burden support redistribution 6% points more than those with lower burden 95% CI = [0.05, 0.06]. Yet, neither of these associations hold up once we purge omitted variable bias. We find similar, although substantively less concerning patterns for system support and extreme discontent. Cross-sectional models tend to overestimate the strength of these associations (by 1.7–4.9 times, or 1–11 percentage points), although they all point in the same direction as our within-respondent estimates.

*Do our results hinge on the inclusion of survey weights*? No. As a final robustness tests, in Figure OA19 in S1 File we demonstrate that our estimates are not sensitive to the inclusion of survey weights.

## Discussion and conclusions

Overall, we find that system support eroded during the COVID-19 crisis in a diverse set of four Western democracies (with the exception of support for democracy). Within-individual analyses show that this erosion could be related to feelings of pandemic fatigue: when respondents' subjective burden of COVID-19 and feelings of anomie grew, they also expressed less support for and satisfaction with the political system they live in. To make things worse, these frustrations appear to spill over to expressions of extreme discontent. When the burden and anomie grew, respondents were more likely to feel a need for chaos, to believe and share misinformation, and to endorse populist ideas. Our findings stand in contrast with prior research demonstrating initial surges to political trust and system support [11, 12], but they are congruent with findings which show that rally-around-the-flag effects decay fast and depend on the (perceived) performance of the authorities [14, 15].

We find no evidence that social solidarity decreased during the initial year of the pandemic, nor that individual-level changes in the burden of the crisis cause systematic shifts in

horizontal social relationships. Why did people link pandemic burden to system support, and extreme discontent, but not to any of the diverse set of outcomes we group under social solidarity? One potential explanation may be that in the initial phases of the pandemic, prior to the roll out of vaccines, governments and health authorities took center stage. Most prominently, they bore full responsibility in introducing or not introducing, lifting or not lifting public and social measures, including, among others, lockdowns, school and workplace closures, bans on public gatherings and mask mandates. Most of our participants have likely experienced the most striking (if well justified) "intrusions" into their lives by the state. This may have overshadowed the role and responsibility of fellow individuals in contributing to viral spread, and thus the burden of the pandemic. There is some evidence that as vaccines were rolled out and it became clear that vaccine hesitancy prevents most societies from reaching sufficient levels of immunity to ease many public and social measures, horizontal relationships were also strained [23].

While there is a clear association between individual-level changes in system support and feelings of anomie and COVID-19 burden, respectively, one particular concerning empirical pattern obtained is that there is a mismatch between the associated aggregate trends. System support decayed constantly over time in all four countries. COVID-19 burden and anomie, in contrast, eased slightly in the summer, most likely due to the improvements in epidemic severity, and restrictions. This disjunction suggests that the pandemic's damage to system support is more permanent or, at least, that system support may not immediately bounce back once the burden of the pandemic lessens.

These conclusions notwithstanding, our analyses are subject to a number of limitations. First, we designed our study in the first weeks of the pandemic, under considerable uncertainty. Accordingly, our design strategy was to cast a wide net and investigate a diverse set of social and political attitudes, which we could benchmark to pre-pandemic levels, yielding a data-driven analysis. Caution is warranted when generalizing to attitudes beyond those measured in our surveys. Although the outcomes we group together show reasonable consistency, it is possible that some other attitudes deviate from these patterns. Perhaps some forms of social solidarity–for example, those related to moralistic condemnation [33]–still suffered from the COVID-19 crisis. Conversely, perhaps some forms of system support showed more resilience than those we measured. Yet, we believe our study offers an important overview of the socio-political effects of the pandemic's first phase and could serve as a starting point for more narrow, confirmatory studies probing specific relationships.

Another limitation due to designing the study early in the pandemic is that our pre-pandemic benchmarks are imperfect. Any changes between the latest waves of the Values Surveys and our wave 1 data are likely a mix of real changes in attitudes and noise from different respondents and survey platforms. We believe our statistical adjustments and the consistency of time trends between the benchmark and our own waves lend some credibility to our findings. Nonetheless, they should be interpreted with caution.

Second, our data are limited in time and space. We study four countries, all in the West, all democratic (even if some of them are backsliding). We believe that the cross-country consistency of our key findings warrants careful generalization to other Western democracies. Still, our results cannot speak to several other regions of the World, where the burden of the pandemic is often heavier, and political system is often more volatile and the fabric of society is frailer. Besides, our study focused on the first phase of the pandemic, before the roll out of vaccines, when government mandated interventions were the primary tool for containing the virus. Thus, it remains mute on the effects of pandemic burden on system support and social solidarity in later stages, as societies sought to vaccinate enough people throughout 2021, or as they were opening up after the omicron wave in the early months of 2022. Still, we believe our

study covers a period of utmost interest, not just because it captures the initial shock of the pandemic, but also because it could be informative for future crises, where an equivalent technological breakthrough may be impossible or may take longer to develop.

Finally, while the panel setup goes a long way in guarding against unobserved confounders, our estimates regarding the effects of anomie and covid burden are still vulnerable to, for example, time variant confounders. Our design sought to balance ecological and internal validity. Future research could manipulate burden and anomie experimentally to estimate unbiased causal effects. Our findings should be useful in benchmarking these estimates to realistic shifts in the predictors.

What are the implications of our findings? We demonstrate that the COVID-19 pandemic is a *total crisis*, affecting many domains of life beyond health and consequently straining the relationships between citizens and the state. This is worrisome first because such factors may make the mitigation of the pandemic itself more difficult: trust in the authorities is among the strongest predictors of willingness to take a vaccine, and several democratic countries struggled to reach high levels of vaccination [42, 43]. But our results are also concerning in a long-term perspective. Hopefully, the anti-system sentiments and extreme discontent subsided once the pandemic was over. Democratic political systems have been facing challenges from populist leaders, growing political polarization, increasing inequalities even before the pandemic. Addressing these social challenges may become even more difficult due to the pandemic. Finally, while the COVID-19 pandemic is arguably the largest global crisis in most of our lifetimes, it may not be the last one. Health experts warn that in an increasingly globalized world pandemics are an increasing threat. Meanwhile, it is still unclear if and how a climate catastrophe could be averted, thus the next total crisis may be looming just over the horizon. Given the impact of the pandemic documented in this manuscript, such crises may become even more difficult to manage.

## Supporting information

**S1 File.**
(PDF)

## Author Contributions

**Conceptualization:** Alexander Bor, Michael Bang Petersen.

**Data curation:** Alexander Bor, Frederik Jørgensen, Michael Bang Petersen.

**Funding acquisition:** Michael Bang Petersen.

**Methodology:** Alexander Bor, Frederik Jørgensen, Michael Bang Petersen.

**Project administration:** Alexander Bor.

**Software:** Alexander Bor.

**Supervision:** Michael Bang Petersen.

**Visualization:** Alexander Bor.

**Writing – original draft:** Alexander Bor, Michael Bang Petersen.

**Writing – review & editing:** Alexander Bor, Frederik Jørgensen, Michael Bang Petersen.

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
