## [Decision Letter · Decision Letter 0]

17 Jan 2023

PONE-D-22-31827The COVID-19 Pandemic Eroded System Support But Not Social SolidarityPLOS ONE

Dear Dr. Bor,

Thank you for submitting your manuscript to PLOS ONE. After careful consideration, we feel that it has merit but does not fully meet PLOS ONE’s publication criteria as it currently stands. Therefore, we invite you to submit a revised version of the manuscript that addresses the points raised during the review process.

We look forward to receiving your revised manuscript.

Kind regards,

Jean-François Daoust

Academic Editor

PLOS ONE

Journal Requirements:

Additional Editor Comments:

Both reviewers provide many comments and suggestions that, I believe, are fair, constructive, and helpful to improve the manuscript. 

R1 provides many comments related to the empirical analyses and the claims made in the paper. Among other things, the issue of the timing of the surveys seems important (see R2’s second point and implication in their comment #3). See also their comment #5 which also relates to one of R1’s minor comment about the descriptive statistics and attrition rates.

R2 believes that the manuscript is somewhat undertheorized, especially when it comes to the concepts measured by the core IV and DV. The question raised is, among others, 'How does your measurement capture those concepts?’ I expect engagement with theoretical work, but given the nature of the question, it will necessarily be intertwined with discussions about the measures/survey data. 

Reviewers' comments:

Reviewer's Responses to Questions

**Comments to the Author**

1. Is the manuscript technically sound, and do the data support the conclusions?

Reviewer #1: Partly

Reviewer #2: Yes

2. Has the statistical analysis been performed appropriately and rigorously? 

Reviewer #1: Yes

Reviewer #2: Yes

3. Have the authors made all data underlying the findings in their manuscript fully available?

Reviewer #1: Yes

Reviewer #2: Yes

4. Is the manuscript presented in an intelligible fashion and written in standard English?

Reviewer #1: Yes

Reviewer #2: Yes

5. Review Comments to the Author

Reviewer #1: I reviewed the paper “The COVID-19 Pandemic Eroded System Support but Not Social Solidarity.” In this paper, the authors report the results of a panel survey conducted in Denmark, Hungary, Italy, and the US over the course of 2020 as the pandemic unfolded. They show that, over three survey waves, respondents are increasingly less satisfied with the way political institutions/democracy work/s in their country (contrary to preferences for redistribution and attitudes towards migration that did not change much). Then, they show that such satisfaction is negatively associated with subjective variables captured of how isolated people fell during the pandemic and how of a burden they think this pandemic is for them personally.

I believe the paper should be published in PLOS One. The data are original, as there are not that many cross-country panel surveys conducted during the COVID-19 pandemic. The analysis is professionally executed and transparently described in the text. Finally, the paper is making an interesting contribution by looking at the evolution of various political and social attitudes from April to December 2020. As the authors correctly point out in the text, most (if not all) existing papers that look at these attitudes focus on the first wave of the pandemic.

That said, I’ve a few comments that the authors should address before publication. They are all quite minor in the sense that they don’t require (much) extra analysis, but they are nonetheless important.

1. As I said, I find the first part of the analysis, in which the authors analyze the evolution of various political and social attitudes during the pandemic, particularly interesting. What is missing though is a measure of whether these changes are statistically significant. Relatedly, Figure 1 could include a confidence interval around each point estimate (not just around those of the pooled variable).

2. From Figure 1, it also appears that the changes between April and December 2020 concern only handful of variables. At various points in the text (including the title), the authors claim that the “system support” variables decrease over this period. In reality, it’s “only” a few of them, namely satisfaction with the way political institutions/democracy work/s (and also whether people feel like proud citizens, but I’m not sure whether this variable is related to the two others). The authors need to be more precise about what attitudes decrease during the panel survey in the abstract, introduction, and conclusion.

3. At various points in the text, the authors make the point that these “system support” variable had already decreased in April 2020 (i.e., the first wave of their panel survey). To back this claim, they compare the variables in their survey to the same variables in the European and World Values Surveys. This is problematic for at least two reasons. First, the people who responded to the authors’ survey are not the same as those who responded to the values surveys. The variables would then only be comparable if the two surveys were randomly sampling the same population. This is a very unlikely given that the survey modes are different (online vs face-to-face). I know that the authors use entropy matching to address this issue, but it’s only a partial fix given that there might still be some unobserved differences between the two samples. Second, I suppose the values surveys were conducted some time before the start of the pandemic, so it’s not clear whether the observed decrease in the variables of interest is due to the pandemic or any event that happened before it.

For these two reasons, I suggest that the authors remove from the text the argument according to which levels of “system support” already decreased in April 2020. They might say, in the results section, that the data suggest the existence of such decrease, but they do not have the data to make it a solid finding that would then appears in the abstract, introduction and conclusion. We simply do not know whether the trend before April 2020.

4. In the second part of the analysis, the authors regress the same social and political attitudes on variables capturing subjective assessments regarding how much a burden the pandemic has been for respondents and how isolated they have felt. They then use two-way fixed effects so that the estimates capture within-respondents and within-waves associations. As I said, the analysis is professionally executed. However, I’m afraid the results cannot be interpreted as “effects”. A key issue is that both the independent and dependent variables are self-reported attitudes. Whereas the two-way fixed effects minimize the possibility that the effect is confounded, it does not address the issue of reverse causality. For example, the authors argue that it’s because respondents fell that the pandemic was a burden for them that their satisfaction with political institutions decreased. But it might be the other way round: Respondents fell that the pandemic was a burden for them because they were unhappy with political institutions. For example, we all know people who were very pessimistic about the impact of the pandemic on their situation (including personal financial situation) although their situation was objectively good. These were also the people who were the most critical of the government.

For this reason, the authors need to be clear in the text that the results of this second part of the analysis are associations and not effects. In some places in the text, they for example use the word “drive”, which is inappropriate. Furthermore, they need to tamper the argument that panel surveys are excellent tools to estimate causal effects especially when one regresses attitudes on attitudes. On p.12, they for example say that that they “constitute a good (perhaps the best) compromise balancing internal validity and generalizability.” This might be true, but that doesn’t mean that they can interpret the regression coefficients as effects. On pp.17-18, they argue that they are “unbiased estimates of causal effects as long as the parallel trends assumption holds”. That’s not correct given that showing the existence of parallel trends does not shut down the possibility of reverse causality.

To be clear: I don’t mean to diminish the value of the paper. I think it’s equally important to show associations between variables. Not every paper needs to have a causal identification strategy. But it’s important to interpret the results for what they are not to misled the reader.

5. A key issue with panel surveys is attrition. Many of the respondent who agree to respond to the first Wave might not agree to respond to subsequent waves. The authors do report the attrition rate, but they do not show whether this affects the representativeness of the sample. Was the sample in the latest wave still representative of the population? This is an important question to assess the external validity of the results.

6. Finally, and this is very minor, but I find that the legend below Figure 3 could be more explicit. Figure 3 has an unusual as the rows show the dependent variables instead of the independent variable. A clear legend would be useful to avoid any confusion.

Reviewer #2: There’s much to like about this manuscript: It studies a relevant topic, uses original and rich data, analyzes this data in a transparent and convincing way and arrives at compelling results. The presentation is straightforward and the entire research process is being made very transparent, with descriptions of the data collection, ethical approval, operationalization, modeling and results, and full replication materials made available on OSF. Plenty of robustness checks substantiate the core results.

My main criticism is that the manuscript seems somewhat under-theorized. While it does provide a (somewhat cursory) review of the literature as well as two psychological mechanisms for explanation, it does not properly conceptualize any of its core independent or dependent variables. What exactly do you mean by “system support”? What by “social solidarity”? How does your measurement capture those concepts?

As far as the literature review is concerned, I think the recent literature on both the relevance of system support for compliance during the pandemic (Bargain and Aminjonov, 2020; Denemark et al., 2022; Weinberg, 2020) and, even more importantly, the effects of the pandemic on system support and societal relations (Anghel and Schulte-Cloos, 2022; Devine et al., 2020; Erhardt et al., 2022; Erhardt et al., 2023; Esaiasson et al., 2021; Kritzinger et al., 2021; Nielsen and Lindvall, 2021; Rump and Zwiener-Collins, 2021; Schraff, 2021) deserves considerably more attention. At the moment, it feels like the authors primarily use the lit review to cite their own papers rather than to discuss the arguments and findings of others.

A couple of minor points:

- Not sure I agree about using all respondents (even those who dropped out in later waves) for descriptive analysis of outcome variables – what is the rationale for doing so?

- How do your results compare to / can be reconciled with the early rally-around-the-flag effects found by others?

- You mention violations of democratic standards in many countries during the pandemic (p. 3). Could those be an alternative explanation for waning system support?

- In terms of typos, I noticed that “analyses” (as in the plural of “analysis”) is often misspelled as “analyzes”. Also, some special characters in authors’ names aren’t being displayed correctly (e.g., Obradovi´c, Power and Sheehy-Skeffington 2020, p. 4).

References

Anghel V and Schulte-Cloos J (2022) COVID-19-Related Anxieties Do Not Decrease Support for Liberal Democracy. European Journal of Political Research online first.

Bargain O and Aminjonov U (2020) Trust and Compliance to Public Health Policies in Times of Covid-19.

Denemark D, Harper T and Attwell K (2022) Vaccine Hesitancy and Trust in Government: A Cross-National Analysis. Australian Journal of Political Science 57(2): 145–163.

Devine D, Gaskell J, Jennings W, et al. (2020) Trust and the Coronavirus Pandemic: What are the Consequences of and for Trust? An Early Review of the Literature. Political Studies Review online first.

Erhardt J, Freitag M and Filsinger M (2023) Leaving Democracy? Pandemic Threat, Emotional Accounts and Regime Support in Comparative Perspective. West European Politics 46(3): 477–499.

Erhardt J, Freitag M, Wamsler S, et al. (2022) What Drives Political Support? Evidence from a Survey Experiment at the Onset of the Corona Crisis. Contemporary Politics 28(4): 429–446.

Esaiasson P, Sohlberg J, Ghersetti M, et al. (2021) How the Coronavirus Crisis Affects Citizen Trust in Institutions and in Unknown Others: Evidence from ‘the Swedish Experiment’. European Journal of Political Research 60(3): 748–760.

Kritzinger S, Foucault M, Lachat R, et al. (2021) ‘Rally Round the Flag’: The COVID-19 Crisis and Trust in the National Government. West European Politics 44(5-6): 1205–1231.

Nielsen JH and Lindvall J (2021) Trust in Government in Sweden and Denmark during the COVID-19 Epidemic. West European Politics 44(5-6): 1180–1204.

Rump M and Zwiener-Collins N (2021) What Determines Political Trust during the COVID-19 Crisis? The Role of Sociotropic and Egotropic Crisis Impact. Journal of Elections, Public Opinion & Parties 31(S1): 259–271.

Schraff D (2021) Political Trust during the Covid‐19 Pandemic: Rally around the Flag or Lockdown Effects? European Journal of Political Research 60(4): 1007–1017.

Weinberg J (2020) Can Political Trust Help to Explain Elite Policy Support and Public Behaviour in Times of Crisis? Evidence from the United Kingdom at the Height of the 2020 Coronavirus Pandemic. Political Studies online first.

6. PLOS authors have the option to publish the peer review history of their article (what does this mean?). If published, this will include your full peer review and any attached files.

Reviewer #1: No

Reviewer #2: **Yes: **Marlene Mauk

---

## [Author Response · Author response to Decision Letter 0]

27 Jun 2023

please see all responses to reviewers in the Cover letter / Revision Memo.

---

## [Decision Letter · Decision Letter 1]

2 Jul 2023

The COVID-19 Pandemic Eroded System Support But Not Social Solidarity

PONE-D-22-31827R1

Dear Dr. Bor,

We’re pleased to inform you that your manuscript has been judged scientifically suitable for publication and will be formally accepted for publication once it meets all outstanding technical requirements.

Kind regards,

Jean-François Daoust

Academic Editor

PLOS ONE

Additional Editor Comments (optional):

Reviewers' comments:

Reviewer's Responses to Questions

**Comments to the Author**

1. If the authors have adequately addressed your comments raised in a previous round of review and you feel that this manuscript is now acceptable for publication, you may indicate that here to bypass the “Comments to the Author” section, enter your conflict of interest statement in the “Confidential to Editor” section, and submit your "Accept" recommendation.

Reviewer #1: All comments have been addressed

Reviewer #2: All comments have been addressed

2. Is the manuscript technically sound, and do the data support the conclusions?

Reviewer #1: Yes

Reviewer #2: Yes

3. Has the statistical analysis been performed appropriately and rigorously? 

Reviewer #1: Yes

Reviewer #2: Yes

4. Have the authors made all data underlying the findings in their manuscript fully available?

Reviewer #1: Yes

Reviewer #2: Yes

5. Is the manuscript presented in an intelligible fashion and written in standard English?

Reviewer #1: Yes

Reviewer #2: Yes

6. Review Comments to the Author

Reviewer #1: I'd like to thank the authors for fully engaging with my comments. This is a great paper. I'm happy to see it published.

Reviewer #2: Thank you for addressing my comments. I am now happy with the manuscript and look forward to seeing it published.

7. PLOS authors have the option to publish the peer review history of their article (what does this mean?). If published, this will include your full peer review and any attached files.

Reviewer #1: **Yes: **Damien Bol

Reviewer #2: No

---

## [Editor Report · Acceptance letter]

8 Aug 2023

PONE-D-22-31827R1 

The COVID-19 Pandemic Eroded System Support But Not Social Solidarity 

Dear Dr. Bor:

I'm pleased to inform you that your manuscript has been deemed suitable for publication in PLOS ONE. Congratulations! Your manuscript is now with our production department. 

Kind regards, 

on behalf of

Dr. Jean-François Daoust 

Academic Editor

PLOS ONE